# Geo-SIC: Learning Deformable Geometric Shapes in Deep Image Classifiers

**Jian Wang**
Computer Science
University of Virginia
`jw4hv@virginia.edu`

**Miaomiao Zhang**
Computer Science & Electrical Computer Engineering
University of Virginia
`mz8rr@virginia.edu`

## Abstract

Deformable shapes provide important and complex geometric features of objects presented in images. However, such information is oftentimes missing or under-utilized as implicit knowledge in many image analysis tasks. This paper presents Geo-SIC, the first deep learning model to learn deformable shapes in a deformation space for an improved performance of image classification. We introduce a newly designed framework that (i) simultaneously derives features from both image and latent shape spaces with large intra-class variations; and (ii) gains increased model interpretability by allowing direct access to the underlying geometric features of image data. In particular, we develop a boosted classification network, equipped with an unsupervised learning of geometric shape representations characterized by diffeomorphic transformations within each class. In contrast to previous approaches using pre-extracted shapes, our model provides a more fundamental approach by naturally learning the most relevant shape features jointly with an image classifier. We demonstrate the effectiveness of our method on both simulated 2D images and real 3D brain magnetic resonance (MR) images. Experimental results show that our model substantially improves the image classification accuracy with an additional benefit of increased model interpretability. Our code is publicly available at `https://github.com/jw4hv/Geo-SIC`

## 1 Introduction

Deformable shapes have been identified to aid image classification for decades, as they capture geometric features that describe changes and variability of objects with complex structures from images (5; 18; 26). Bountiful literature demonstrates that the robustness of shapes to variations in image intensity and texture (e.g., noisy or corrupted data) makes it a reliable cue for image analysis tasks (31; 26; 43). For example, abnormal shape changes of anatomical structures are strong predictors of serious diseases and health problems, e.g., brain shrinkage caused by neurodegenerative disorders (16; 49), irregular heart motions caused by cardiac arrhythmia (3; 8), and pulmonary edema caused by infectious lung diseases (42; 44). Existing methods have studied various representations of geometric shapes, including landmarks (7; 9; 14), point clouds (1), binary segmentations (10; 38), and medial axes (32). A very recent research area in geometric deep learning (11; 33) has investigated mathematical representations of shapes in the form of analytic graphs or points and then uses them to synthesize shapes. These aforementioned techniques often ignore objects' interior structures; hence do not capture the intricacies of complex objects in images. In contrast, deformation-based shape representations (based on elastic deformations or fluid flows) focus on highly detailed shape information from images (13; 35). With the underlying assumption that objects in many generic classes can be described as deformed versions of an ideal template, descriptors in this class arise naturally by matching the template to an input image. This procedure is also known as *atlas building* (22; 40; 48). The resulting transformation is then considered a shape that reflects geometric

36th Conference on Neural Information Processing Systems (NeurIPS 2022).

changes. In this paper, we will feature deformation-based shape representations that offer more flexibility in describing shape changes and variability of complex structures. However, our developed framework can be easily adapted to other types of representations, including those characterized by landmarks, binary segmentations, curves, and surfaces.

Inspired by the advantages of incorporating shape information in image analysis tasks, current deep learning-based classification networks have been mostly successful in using pre-extracted shapes from images (45; 29; 7). However, these methods require preprocessed shape data and oftentimes achieve a suboptimal solution in identifying shape features that are most representative to differentiate different classes of images. An explicit learning of deformable shapes in deep image classifiers has been missing. This limits the power of classification models where quantifying and analyzing geometric shapes is critical.

In this paper, we introduce a novel deep learning image classification model, named as Geo-SIC, that jointly learns deformable shapes in a multi-template deformation space. More specifically, Geo-SIC provides an unsupervised learning of deformation-based shape representations via a newly designed sub-network of atlas building. Different from previous deep learning based atlas building approaches (15; 20), we employ an efficient parameterization of deformations in a low-dimensional Fourier space (46) to speed up the training inference. The major contribution of Geo-SIC is three folds:

(i) In contrast to previous approaches treating shapes as preprocessed objects from images, Geo-SIC provides a more fundamental approach by merging shape features naturally in the learning process of classification. To the best of our knowledge, Geo-SIC was the first to learn deformation-based shape descriptors within an image classifier. It provides an image distance function of both intensity and geometric changes that are most relevant to classify different groups.

(ii) Geo-SIC performs a simultaneous feature extraction from both image and learned shape spaces. With these integrated features, Geo-SIC achieves an improved accuracy and robustness of image classification. An additional benefit of Geo-SIC is increased model interpretability because of its access to the underlying geometric features of image data.

(iii) Geo-SIC provides an efficient geometric learning network via atlas building in a compact and low-dimensional shape space. This reduces the computational complexity of model training in atlas building, especially for high-dimensional image data (i.e., 3D brain MRIs).

We demonstrate the effectiveness of Geo-SIC on both synthetic 2D images and real 3D brain MR images. Experimental results show that our model substantially improves the classification accuracy compared to a wide variety of models without jointly learned geometric features. We then visualize the class activation maps by using gradient-weighted class activation mapping (Grad-CAM) (36). The highlighted regions show that Geo-SIC attracts more attention to geometric shape features that positively contribute to the accuracy of classifiers.

## 2 Background: Deformation-based Shape Representations via Atlas Building

This section briefly reviews the concept of atlas building (22), which is commonly used to derive deformation-based shape representations from images. With the underlying assumption that the geometric information in the deformations conveys a shape, descriptors in this class arise naturally by matching a template to an input image with smoothness constraints on the deformation field.

Given a number of $N$ images $\{I_1, \cdots, I_N\}$, the problem of atlas building is to find a mean or template image $I$ and deformation fields $\phi_1, \cdots, \phi_N$ that minimize the energy function

$$E(I, \phi_n) = \sum_{n=1}^{N} \frac{1}{\sigma^2} \text{Dist}[I \circ \phi_n^{-1}, I_n] + \text{Reg}(\phi_n), \qquad (1)$$

where $\sigma^2$ is a noise variance and $\circ$ denotes an interpolation operator that deforms image $I$ with an estimated transformation $\phi_n$. The $\text{Dist}(\cdot, \cdot)$ is a distance function that measures the dissimilarity between images, i.e., sum-of-squared differences (6), normalized cross correlation (4), and mutual information (41). The $\text{Reg}(\cdot)$ is a regularization that guarantees the smoothness of transformations.

## 2.1 Shape Representations In The Tangent Space Of Diffeomorphisms

In many applications, it is natural to require the deformation field to be a diffeomorphism, i.e., a differentiable, bijective mapping with a differentiable inverse. Shape representations in the space of diffeomorphic transformations highlight a set of desirable features: (i) they capture large geometric variations within image groups; (ii) the topology of objects in the image remains intact; and (iii) no non-differentiable artifacts, such as creases or sharp corners, are introduced. Moreover, a theoretical framework of Large Deformation Diffeomorphic Metric Mapping (LDDMM) defines a metric in the space of diffeomorphic transformations that in turn induces a distance metric on the shape space (6).

Given an open and bounded $d$-dimensional domain $\Omega \subset \mathbb{R}^d$, we use $\text{Diff}(\Omega)$ to denote a space of diffeomorphisms (i.e., a one-to-one smooth and invertible smooth transformation) and its tangent space $V = T\text{Diff}(\Omega)$. The LDDMM algorithm (6) provides a distance metric in the deformation-based shape space, which is used as a regularization of atlas building in Eq. 1. Such a distance metric is formulated as an integral of the Sobolev norm of the time-dependent velocity field $v_n(t) \in V (t \in [0, 1])$ in the tangent space, i.e.,

$$\text{Reg}(\phi_n) = \int_0^1 (\mathcal{L}v_n(t), v_n(t))\, dt, \quad \text{with} \quad \frac{d\phi_n^{-1}(t)}{dt} = -D\phi_n^{-1}(t) \cdot v_n(t), \qquad (2)$$

where $\mathcal{L} : V \to V^*$ is a symmetric, positive-definite differential operator that maps a tangent vector $v(t) \in V$ into its dual space as a momentum vector $m(t) \in V^*$. We typically write $m(t) = \mathcal{L}v(t)$, or $v(t) = \mathcal{K}m(t)$, with $\mathcal{K}$ being an inverse operator of $\mathcal{L}$. In this paper, we adopt a commonly used Laplacian operator $\mathcal{L} = (-\alpha\Delta + \text{Id})^3$, where $\alpha$ is a weighting parameter that controls the smoothness of transformation fields and Id is an identity matrix. The $(\cdot, \cdot)$ is a dual pairing, which is similar to an inner product between vectors. The operator $D$ denotes a Jacobian matrix and $\cdot$ represents an element-wise matrix multiplication.

According to a well known geodesic shooting algorithm (39), the minimum of Eq. (2) is uniquely determined by solving a Euler-Poincaré differential equation (EPDiff) (2; 25) with a given initial condition. This nicely proves that the deformation-based shape descriptor $\phi_n$ can be fully characterized by an initial velocity field $v_n(0)$, which lies in the tangent space of diffeomorphisms. A recent research further identifies an equivalent but more efficient way to reparameterize these initial velocity fields in a low-dimensional bandlimited space (46; 47).

Let $\widetilde{\text{Diff}}(\Omega)$ and $\tilde{V}$ denote the bandlimited space of diffeomorphisms and velocity fields respectively. The EPDiff is reformulated in a complex-valued Fourier space with much lower dimensions, i.e.,

$$\frac{\partial \tilde{v}_n(t)}{\partial t} = -\tilde{\mathcal{K}}\left[(\tilde{\mathcal{D}}\tilde{v}_n(t))^T \star \tilde{\mathcal{L}}\tilde{v}_n(t) + \tilde{\nabla} \cdot (\tilde{\mathcal{L}}\tilde{v}_n(t) \otimes \tilde{v}_n(t))\right], \qquad (3)$$

where $\star$ is the truncated matrix-vector field auto-correlation. Here $\tilde{\mathcal{K}}$ is an inverse operator of $\tilde{\mathcal{L}}$, which is the Fourier transform of a Laplacian operator $\mathcal{L}$. The operator $\tilde{\mathcal{D}}$ represents the Fourier frequencies of a Jacobian matrix with central difference approximation, and $*$ is a circular convolution with zero padding to avoid aliasing [1]. The operator $\tilde{\nabla}\cdot$ is the discrete divergence of a vector field, and $\otimes$ represents a tensor product between Fourier frequencies.

We are now ready to optimize the problem of atlas building (Eq. (1)) with reduced computational complexity in a low-dimensional bandlimited space as

$$E(I, \phi_n) = \sum_{n=1}^{N} \frac{1}{\sigma^2} \text{Dist}[I \circ \phi_n^{-1}, I_n] + (\mathcal{L}\tilde{v}_n(0), \tilde{v}_n(0)), \quad \text{s.t. Eq. (2)\& (3)}. \qquad (4)$$

The deformation $\phi_n^{-1}$ corresponds to $\tilde{\phi}_n^{-1}$ in Fourier space via the Fourier transform $\mathcal{F}(\phi_n^{-1}) = \tilde{\phi}_n^{-1}$, or its inverse $\phi_n^{-1} = \mathcal{F}^{-1}(\tilde{\phi}_n^{-1})$. Note that we will drop the time index $t$, i.e., $\tilde{v}_n(0) \triangleq \tilde{v}_n$, for simplified notations in next sections.

---

[1]To prevent the domain from growing infinity, we truncate the output of the convolution in each dimension to a suitable finite set.

## 3  Our Method: Geo-SIC

In this section, we present a novel deep image classifier (Geo-SIC) that explicitly learns geometric shape representations for an improved performance of accuracy, as well as increased model interpretability. Geo-SIC consists of two modules: an unsupervised learning of geometric shapes via an atlas building network, and a boosted classification network that integrates features from both images and learned shape spaces. Details of our network architecture are introduced as follows.

**Geometric shape learning based on an atlas building network.** Let $(\theta_g^E, \theta_g^D)$ be the parameters of an encoder-decoder in our geometric learning network. Consider a number of $J$ image classes, there exists a number of $N_j, j \in \{1, \ldots, J\}$ images in each class. Our atlas building network will learn the shape representations, also known as initial velocity fields $\tilde{v}_n(\theta_g^E, \theta_g^D), n \in \{1, \cdots, N_j\}$, with an updated atlas $I_j$. We adopt the architecture of UNet (34) in this work, however, other network structures such as UNet++ (50) and TransUNet (12) can be easily applied.

**Boosted image classification network.** Let $\theta_c^E$ be the parameters of an encoder that extracts features from image spaces. We develop a feature fusion module that integrates geometric shape and image features in a latent space parameterized by $\theta_{gc}(\theta_g^E, \theta_c^E)$. This boosted classification network will predict a class label $y_{nj}(\theta_{gc})$ for each input image.

**Network loss.** The loss function of Geo-SIC includes the loss from both geometric learning and classification network. Given a set of image class labels $\hat{y}_{nj}$, we define $\Theta = (\theta_g^E, \theta_g^D, \theta_c^E, \theta_{gc})$ for all network parameters and formulate the total loss of Geo-SIC as

$$l(\Theta) = \sum_{n=1}^{N_j} \sum_{j=1}^{J} [\frac{1}{\sigma_j^2} \|I_j \circ \phi_{nj}^{-1} \left( \tilde{v}_{nj}(\theta_g^E, \theta_g^D) \right) - I_{nj}\|^2 + (\tilde{\mathcal{L}}_j \tilde{v}_{nj}(\theta_g^E, \theta_g^D), \tilde{v}_{nj}(\theta_g^E, \theta_g^D))$$
$$- \lambda \hat{y}_{nj} \cdot \log y_{nj}(\theta_{gc})] + \text{reg}(\Theta), \quad \text{s.t.} \quad \text{Eq. (2)\& (3)}, \tag{5}$$

where $\text{reg}(\cdot)$ is a regularity term on the network parameters.

An overview of our proposed Geo-SIC network is shown in Fig. 1.

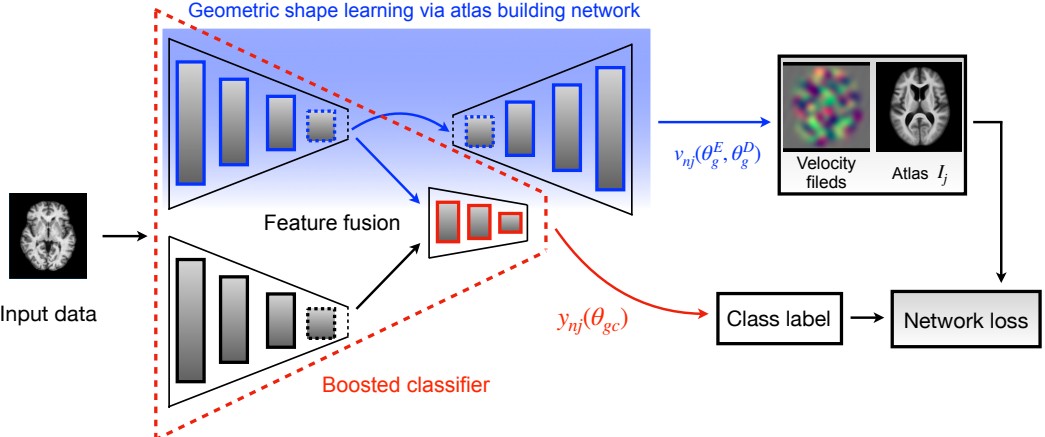

Figure 1: An overview of our proposed Geo-SIC network.

### 3.1  Alternating Optimization For Network Training

We develop an alternating optimization scheme (30) to minimize the network loss defined in Eq. (5). All parameters are optimized jointly by alternating between the training of geometric shape learning and image classification networks.

**Training of geometric shape learning via atlas building.** In contrast to current approaches that parameterize deformation-based shapes in a high-dimensional image space (20; 15), our model employs an efficient reparameterization in a low-dimensional bandlimited space (46). This makes the computation of geodesic constraints (Eq. (2) and Eq. (3)) required in the loss function substantially faster in each forward/backward propagation during the training process. The loss of training in the sub-module of atlas building network is

$$
l_{\text{Geo}}(\theta_g^E, \theta_g^D, I_j) = \sum_{n=1}^{N_j} \sum_{j=1}^{J} [\frac{1}{\sigma_j^2} \| I_j \circ \phi_{nj}^{-1}\left( \tilde{v}_{nj}(\theta_g^E, \theta_g^D) \right) - I_{nj} \|^2 + (\tilde{\mathcal{L}}_j \tilde{v}_{nj}(\theta_g^E, \theta_g^D), \tilde{v}_{nj}(\theta_g^E, \theta_g^D))
$$
$$
+ \operatorname{reg}(\theta_g^E, \theta_g^D), \quad s.t. \quad \text{Eq. (2)\& (3)}, \tag{6}
$$

Similar to (20; 15), we treat atlas $I_j$ as a network parameter and update it accordingly. To guarantee the network optimization stays in the tangent space of diffeomorphisms, we pull back the network gradient with regard to initial velocity fields by backward integrating adjoint jacobi fields (46) each time after the forward pass. More details are included in the supplementary materials.

**Training of boosted image classifier.** As highlighted in the red box in Fig. 1, Geo-SIC extracts both images and geometric features and then integrates them into a feature fusion block. This boosted classifier is optimized over the loss defined as

$$
l_{\text{SIC}}(\theta_{gc}) = -\lambda \sum_{n=1}^{N_j} \sum_{j=1}^{J} \hat{y}_{nj} \cdot \log y_{nj} \left( \theta_{gc}(\theta_g^E, \theta_c^E) \right) + \operatorname{reg}\left( \theta_{gc}(\theta_g^E, \theta_c^E) \right). \tag{7}
$$

A summary of our joint learning of Geo-SIC through an alternating optimization is in Alg. 1.

---

**Algorithm 1:** Joint learning of Geo-SIC.

---

**Input** : A set of images $\{I_{nj}\}$ with class labels $\{\hat{y}_{nj}\}$ and a number of iterations $r$.
**Output :** Predicted class label, atlas, and initial velocity fields.

1 **for** $i = 1$ to $r$ **do**
    /* Train geometric learning network                                    */
2      Minimize the atlas building loss in Eq. (6);
3      Output the atlases $\{I_j\}$ and predicted initial velocity fields $\{v_{nj}\}$ for all image classes;
    /* Train the boosted image classifier                                   */
4      With learned shape features, Minimizing the boosted classification loss in Eq. (7);
5      Output the predicted class labels $\{\hat{y}_{nj}\}$;
6 **end**
7 **Until** convergence

---

## 4 Experimental Evaluation

### 4.1 Data

**2D data set.** We choose 50000 images (including five classes, circle, cloud, envelope, square, and triangle are shown in Fig 3) of Google Quickdraw dataset (21), a collection of categorized drawings contributed by online players in a drawing game. We run affine transformation within each class as prepossessing and upsample each image from $28 \times 28$ to $100 \times 100$.

**3D brain MRI.** For brain data, we include 373 public T1-weighted brain MRI scans from Open Access Series of Imaging Studies (OASIS) (17). All 150 subjects are aged from 60 to 96 with Alzheimer's disease diagnosis (79 cases for dementia and 71 cases for non-demented). All MRIs were all pre-processed as $256 \times 256 \times 256$, $1.25mm^3$ isotropic voxels, and underwent skull-stripped, intensity normalized, bias field corrected, and pre-aligned with affine transformation.

## 4.2 Experiments

**Classification evaluation.** We demonstrate the effectiveness of our model on both 2D synthetic data and 3D brain MRI scans. We select four classification backbones (AlexNet (23), a five-block 3D CNN, ResNet18 (19), and VGG19 (37)) as baseline methods. For CNN, we use a 3D convolutional layer with a $5 \times 5 \times 5$ convolutional kernel size, a batch normalization (BN) layer with activation functions (PReLU or ReLU), and a $2 \times 2 \times 2$ 3D max-pooling in each CNN block. For a fair comparison, we show the results of Geo-SIC by replacing the backbone in our model with all baselines (named as Geo-SIC:Alex, Geo-SIC:CNN, Geo-SIC:Res, and Geo-SIC:VGG).

To further investigate the advantages of our joint learning, we compare with two-step approaches (Two-step Alex, Two-step CNN, Two-step Res, and Two-step VGG), where the geometric learning network is treated as a preprocessing step for geometric feature extraction. We report the average accuracy, F1-score, AUC, sensitivity, and specificity for all methods. We also show the receiver operating characteristic (ROC) curves by plotting the true positive rate (TPR) against the false positive rate (FPR) at various threshold settings.

**Robustness and interpretability.** We demonstrate the robustness of Geo-SIC to variations in image intensity by adding different scales of universal adversarial noises (28; 27) in both 2D and 3D images. We adopt an iterative algorithm (27) that computes the universal perturbations to send perturbed images outside of the decision boundary of the classifier while fooling most images without visibly changing image data. We then compare the image classification accuracy for the baseline (selected from the best performance in backbones) and our model.

To better understand the model interpretability in terms of network attention of all models, we visualize the Grad-CAM (36) of the last neural network layer.

**Atlas evaluation.** We also evaluate the performance of our newly designed atlas building network, which serves as a core part of Geo-SIC. We compare the estimated atlas of Geo-SIC that achieves the highest AUC with three atlas-building methods: a diffeomorphic autoencoder (20) (LagoMorph), a deep learning based conditional template estimation method (15) (Con-Temp) and a Bayesian atlas building framework with hyper priors (40) (Hier-Baye).

We report the total computational time and memory consumption on 3D real brain images. To measure the sharpness of estimated atlas $I$, we adopt a metric of normalized standard deviation computed from randomly selected 4000 image patches (24). Given $M(i)$, a patch around a voxel $i$ of an atlas $I$, the local measure of the sharpness at voxel $i$ is defined as $\text{sharpness}(I(i)) = \text{sd}_{M(i)}(I)/\text{avg}_{M(i)}(I)$, where sd and avg denote the standard deviation and the mean of $M_i$.

**Parameter Setting.** We set an optimal dimension of the low-dimensional shape representation as $16^2$ for 2D dataset and $32^3$ for 3D dataset. We set parameter $\alpha = 3$ for the operator $\tilde{\mathcal{L}}$, the number of time steps for Euler integration in EPDiff (Eq. (3)) as 10. We set the noise variance $\sigma = 0.02$. We set the batch size as 16 and use the cosine annealing learning rate schedule that starts from a learning rate $\eta = 1e - 3$ for network training. We run 1000 epochs with the Adam optimizer and save networks with the best validation performance for all models. All networks are trained with an i7, 9700K CPU with 32 GB internal memory. The training and prediction procedure of all learning-based methods are performed on four Nvidia GTX 2080Ti GPUs. For both 2D and 3D datasets, we split the images by using 70% as training images, 15% as validation images, and 15% as testing images.

## 4.3 Results

### 4.3.1 Classification on 2D synthetic images

Table. 1 reports the classification performance across all methods over model accuracy and five micro-averaged evaluation metrics. Our method achieves the highest model accuracy with comparable micro-averaged AUC, F-1 score, precision, sensitivity, and specificity. Compare to all baselines including two-step approaches, Geo-SIC achieves the best classification performance.

Fig. 2 visualizes the micro-averages ROC curve for baselines and our proposed Geo-SIC. For four sets of comparisons, all curves produced by our classifiers are with a larger AUC and are closer to the top-left corner, which indicates a better classification performance.

Table 1: Classification performance comparison on 2D synthetic data over six metrics (micro-average).

| Models | Accuracy | AUC | F-1 score | Precision | Sensitivity | Specificity |
|---|---|---|---|---|---|---|
| AlexNet | 0.880 | 0.931 | 0.876 | 0.914 | 0.880 | 0.920 |
| Two-step AlexNet | 0.891 | 0.955 | 0.891 | 0.906 | 0.891 | 0.905 |
| **Geo-SIC: Alex** | **0.928** | **0.977** | **0.927** | **0.950** | **0.928** | **0.952** |
| CNN | 0.861 | 0.928 | 0.857 | 0.874 | 0.861 | 0.881 |
| Two-step CNN | 0.869 | 0.931 | 0.866 | **0.914** | 0.869 | **0.918** |
| **Geo-SIC: CNN** | **0.897** | **0.954** | **0.898** | 0.911 | **0.897** | 0.910 |
| ResNet18 | 0.875 | 0.941 | 0.877 | 0.885 | 0.875 | 0.883 |
| Two-step Res | 0.911 | 0.972 | 0.912 | 0.935 | 0.911 | 0.883 |
| **Geo-SIC: Res** | **0.935** | **0.983** | **0.928** | **0.948** | **0.935** | **0.956** |
| VGG19 | 0.883 | 0.946 | 0.882 | 0.904 | 0.883 | 0.905 |
| Two-step VGG | 0.895 | 0.951 | 0.888 | 0.909 | 0.895 | 0.919 |
| **Geo-SIC: VGG** | **0.927** | **0.980** | **0.924** | **0.957** | **0.927** | **0.960** |

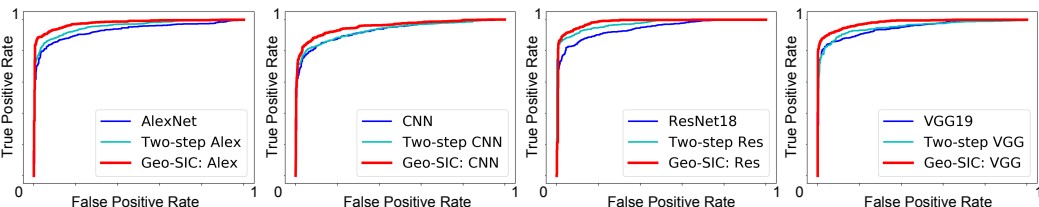

Figure 2: ROC curves of multi-class classification comparison between baselines and the proposed method.

Fig. 3 (left) displays the heat maps overlaid with 2D QuickDraw data for all methods. Geo-SIC produces more explainable heat maps that are geometrically aligned with the original shapes. It indicates that the latent shape space attracts more attention to geometric features that positively contribute to the performance of classifiers. Fig. 3 (right) visualizes a comparison of atlases generated by baselines and Geo-SIC. It shows that Geo-SIC produces image atlases with the best visual quality, e.g., clearer circle/cloud edges, and sharper triangle/square corners.

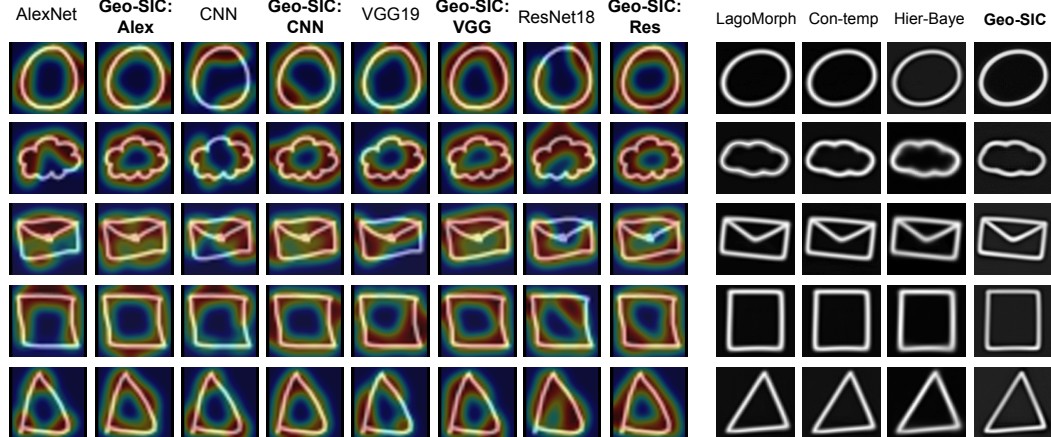

Figure 3: Left: visualization of Grad-CAMs on mutil-atlas building task of five different geometric shapes.; Right: atlas comparison between state-of-the-arts and Geo-SIC.

#### 4.3.2 Classification on 3D real brain MRIs

Table. 2 reports the model performance for brain images. Our method Geo-SIC outperforms all baselines with comparable accuracy, AUC, F1-score, precision, sensitivity, and specificity. It shows

that Geo-SIC substantially improves the classification performance with a reduced misclassification rate.

Table 2: Classification performance comparison on 3D brain images over six metrics.

| Models | Accuracy | AUC | F-1 score | Precision | Sensitivity | Specificity |
|---|---|---|---|---|---|---|
| AlexNet | 0.791 | 0.861 | 0.796 | 0.787 | 0.806 | 0.775 |
| Two-step AlexNet | 0.800 | 0.887 | 0.809 | 0.822 | 0.796 | 0.804 |
| **Geo-SIC: Alex** | **0.835** | **0.917** | **0.827** | **0.823** | **0.831** | **0.839** |
| CNN | 0.779 | 0.860 | 0.773 | 0.804 | 0.744 | 0.814 |
| Two-step CNN | 0.788 | 0.883 | 0.791 | 0.794 | 0.787 | 0.789 |
| **Geo-SIC: CNN** | **0.824** | **0.902** | **0.828** | **0.833** | **0.822** | **0.825** |
| ResNet18 | 0.805 | 0.887 | 0.811 | 0.813 | 0.809 | 0.800 |
| Two-step Res | 0.833 | 0.917 | 0.836 | 0.804 | 0.870 | 0.797 |
| **Geo-SIC: Res** | **0.873** | **0.933** | **0.874** | **0.874** | **0.874** | **0.872** |
| VGG19 | 0.805 | 0.871 | 0.813 | 0.807 | 0.818 | 0.790 |
| Two-step VGG | **0.865** | 0.880 | 0.838 | **0.822** | 0.854 | **0.826** |
| **Geo-SIC: VGG** | 0.852 | **0.930** | **0.850** | 0.820 | **0.881** | 0.825 |

Fig. 4 visualizes the ROC curves for all algorithms. For four sets of comparisons, our boosted classifiers offer curves closer to the top-left corner with higher AUC values. It shows that Geo-SIC has better performance in distinguishing between healthy control and Alzheimer's disease groups.

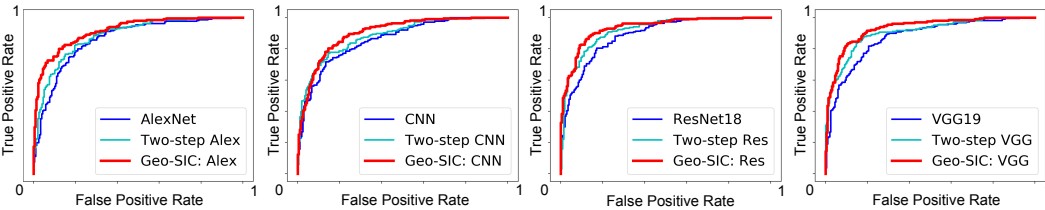

Figure 4: ROC curves of binary classification comparison for baselines, two-step approaches, and Geo-SIC.

### 4.3.3 Robustness and interpretability

Fig. 5 shows that Geo-SIC consistently achieves better classification accuracy ($\sim 10\%$ higher in average ) than baseline algorithms (i.e., VGG19) across different levels of adversarial attacks (i.e., $\epsilon = 5e-3, 5e-2, 5e-1$) on image intensity. This indicates that our model is able to improve the classification model robustness by providing jointly learned geometric features.

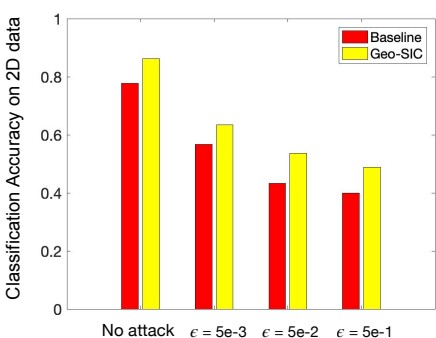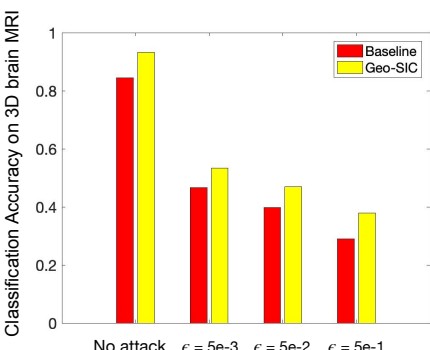

Figure 5: Classification accuracy comparison between baseline and Geo-SIC under different scales of adversarial noise attack for 2D and 3D image classification.

Fig. 6 displays the heat maps (produced by Grad-CAM) overlaid with brain MRIs. Our generated heat maps are fairly aligned with brain structures (e.g., ventricle), which are critical to Alzheimer's disease diagnosis. It profiles the fact that the latent geometric features guide the neural networks to capture the most anatomically meaningful brain regions for distinguishing healthy vs. disease groups. For both dementia and non-dementia cases, the heat maps of Geo-SIC are more explainable. This shows that our model has great clinical potential for better analysis of Alzheimer's disease.

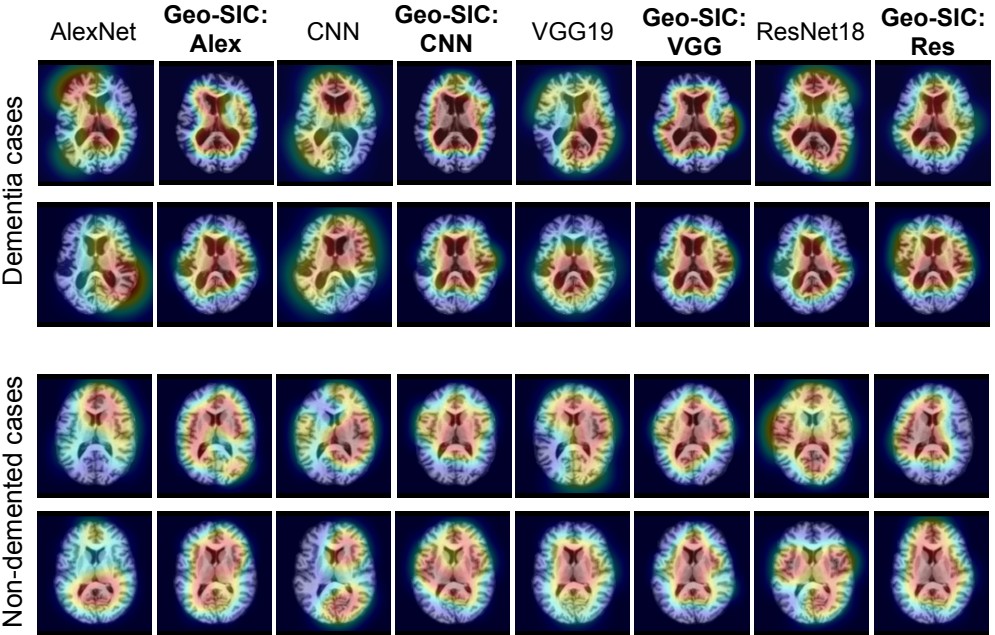

Figure 6: Visualization of Grad-CAMs on single-atlas building of dementia and non-dementia. Left to right: Grad-CAM heatmaps generated by AlexNet, Geo-SIC:Alex, CNN, Geo-SIC:CNN, ResNet18, Geo-SIC:Res, VGG19 and Geo-SIC:VGG.

### 4.3.4 Atlas of 3D images

Fig. 7 (top) visualizes a comparison of the atlas on real brain MRI scans. With the benefits of reparameterizing the deformation fields in a low-dimensional bandlimited space (46), our model obtains better quality of the atlas with sharper details. More specifically, Geo-SIC offers a better brain atlas with clearer anatomical structures, e.g., ventricle, grey and white matter. Fig. 7 (bottom left) quantitatively reports the sharpness metric of all methods. Fig. 7 (bottom right) shows the comparison of computational time and memory consumption across all methods. Although Con-Temp is slightly faster than Geo-SCI due to a very different parameterization of velocity fields (stationary velocity rather than the time-dependent velocity in other methods), it achieves a less sharp atlas and still requires larger memory consumption than our model. Compared with the other methods (Lagomorph and Hier-Baye) that employ time-dependent velocity fields, Geo-SIC substantially reduces time consumption and memory consumption.

## 5   Conclusions & Discussion

This paper presents a novel deep learning model, named as Geo-SIC, that for the first time incorporates deformable geometric shape learning into deep image classifiers. We jointly learn a boosted classifier with an unsupervised shape learning network via atlas building. To achieve this goal, we define a new joint loss function with an alternating optimization scheme. An additional benefit is that Geo-SIC provides efficient shape representations in a low-dimensional bandlimited space. Experimental results on both 2D synthetic data and 3D brain MRI scans show that our model gains an improved classification performance while producing a sharper atlas with better visual quality. In addition,

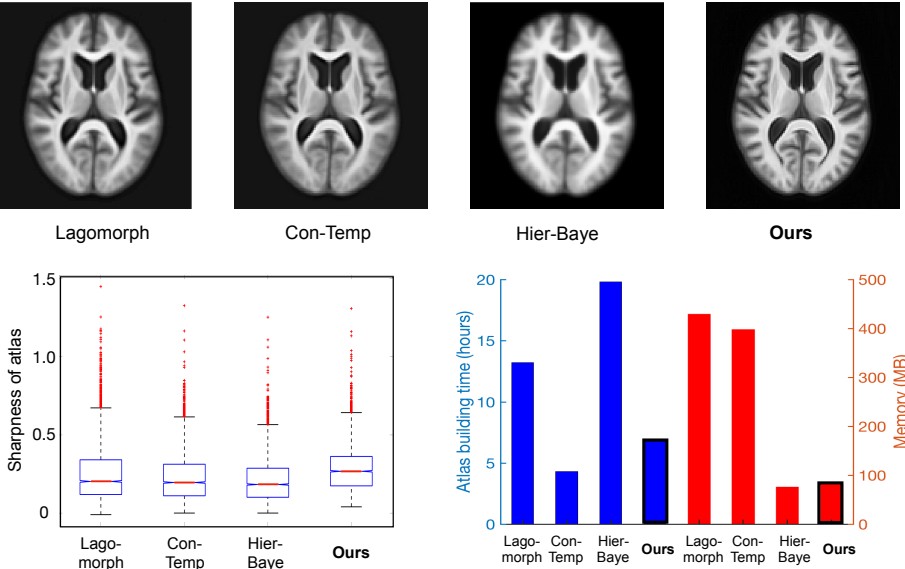

Figure 7: Top: atlas comparison between state-of-the-arts and Geo-SIC. Bottom left: sharpness metric evaluation of atlas (the higher the better).The mean values of the sharpness metric of three baselines and **Geo-SIC** are, 0.259, 0.235, 0.218, and **0.2867**; bottom left: average time and memory consumption comparison for atlas building.

compared with the state-of-the-arts, our model is more explainable in terms of interpreting the network attention on geometric features. The theoretical tools developed in this paper are generic to a wide variety of combinations of shape representations and classification backbones. Geo-SIC not only has a great potential to impact clinically diagnostic routines, such as Alzheimer's disease detection, or post-treatment for patient care, but also bridges the gap between the developed deformable shape learning theories and classification-based applications. Future work to extend our Geo-SIC can be (i) modeling multiple templates within each class to capture multimodal image distributions, and (ii) incorporating images with missing data values that are caused by occlusions, or appearance changes such as tumor growth.

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
