# Supplementary Materials for
# Geo-SIC: Learning Deformable Geometric Shapes in Deep Image Classifiers

**Jian Wang**
Computer Science
University of Virginia
jw4hv@virginia.edu

**Miaomiao Zhang**
Computer Science & Electrical Computer Engineering
University of Virginia
mz8rr@virginia.edu

This section will cover (i) the derivations of pulling back the network gradient into the space of initial velocity fields $\tilde{v}_{nj}$ each time after forward propagation, and (ii) a complexity analysis of our unsupervised atlas building network using the low-dimensional parameterizations.

## 1 Derivations of the gradient for the atlas building network

We recall that the loss of the geometric shape learning based on atlas building network is

$$
l_{\text{Geo}}(\theta_g^E, \theta_g^D, I_j) = \sum_{n=1}^{N_j} \sum_{j=1}^{J} [\frac{1}{\sigma_j^2} \| I_j \circ \phi_{nj}^{-1} \left( \tilde{v}_{nj}(\theta_g^E, \theta_g^D) \right) - I_{nj} \|^2 + (\tilde{\mathcal{L}}_j \tilde{v}_{nj}(\theta_g^E, \theta_g^D), \tilde{v}_{nj}(\theta_g^E, \theta_g^D))
$$
$$
+ \operatorname{reg}(\theta_g^E, \theta_g^D), \quad s.t. \quad \text{Eq.(2)} \& (3).
$$

We set the network output $\tilde{v}_{nj}(\theta_g^E, \theta_g^D)$ as the initial condition of geodesic shooting, and adopt a forward-backward shooting approach (2; 3) that employs adjoint Jacobi fields in Fourier space. With a simplified math notation $\Theta_g = (\theta_g^E, \theta_g^D)$, we derive the gradient of the loss function with respect to the predicted initial velocity fields $\tilde{v}_{nj}$ before back-propagation as follows

(i) Forward integrating the geodesic shooting equation (a.k.a. EPDiff) in Eq.(2) to compute $\tilde{v}_{nj}(\Theta_g)_{t=1}$ at time point $t = 1$ after obtaining the predicted initial velocity fields $\tilde{v}_{nj}$ from network forward-propagation;

(ii) Compute the gradient of the loss function $l_{\text{Geo}}(\Theta_g, I_j)$ with respect to $\tilde{v}_{nj}(\Theta_g)_{t=1}$,

$$
\nabla_{\tilde{v}_{nj}(\Theta_g)_{t=1}} l_{\text{Geo}} = \tilde{\mathcal{K}} \left( \frac{1}{\sigma_j^2} (I_j \circ \phi_n^{-1} [\tilde{v}_{nj}(\Theta_g)_{t=1}] - I_{nj}) \cdot \nabla(I_j \circ \phi_{nj} [\tilde{v}_{nj}(\Theta_g)_{t=1}]) \right);
$$

(iii) Bring the gradient in (ii) back to the space of initial velocity fields defined at the time point $t = 0$ by integrating adjoint Jacobi fields backward in time obtain $\nabla_{\tilde{v}_{nj}(\Theta_g)} l_{\text{Geo}}$,

$$
\frac{d\hat{v}}{dt} = -\operatorname{ad}_{\tilde{v}}^{\dagger} \hat{u}, \quad \frac{d\hat{u}}{dt} = -\hat{v} - \operatorname{ad}_{\tilde{v}} \hat{u} + \operatorname{ad}_{\hat{u}}^{\dagger} \tilde{v},
$$

where $\hat{v} \in V$ are introduced adjoint variables with an initial condition $\hat{u} = 0, \hat{v} = \nabla_{\tilde{v}_n(\Theta_g)_1} l_{\text{Geo}}$ at $t = 1$. Here $\operatorname{ad}^{\dagger}$ is an adjoint operator to the negative Lie bracket of vector fields, $\operatorname{ad}_{\tilde{v}} \tilde{w} = -[\tilde{v}, \tilde{w}] = \tilde{\mathcal{D}} \tilde{v} * \tilde{w} - \tilde{\mathcal{D}} \tilde{w} * \tilde{v}$.

## 2 Computational complexity analysis

In our framework, optimizing the loss of atlas building network with a low-dimensional geodesic constraint Eq. (3) is significantly faster than solving Eq. (1) in high-dimensional image space. We list

36th Conference on Neural Information Processing Systems (NeurIPS 2022).

out the details of computational complexity for geodesic shooting of Geo-SIC and compare it with Lagomorph (1) (a deep atlas learning approach using LDDMM in the spatial domain) in Table. 1.

Table 1: Computational complexity of batchwise geodesic shooting of Geo-SIC and Lagomorpch ($T$: number of integration time steps; $d$: image dimension; $q$: number of low frequencies along each dimension; $Q$: number of image voxels along each dimension; $B$: batch size.)

|  | Complexity | | Memory | |
|---|---|---|---|---|
|  | Geo-SIC | Lagomorph | Geo-SIC | Lagomorph |
| (i). Forward shooting | $\mathcal{O}(BTq^d \log q)$ | $\mathcal{O}(BTQ^d \log Q)$ | $\mathcal{O}(BTq^d)$ | $\mathcal{O}(BTQ^d)$ |
| (ii). Compute gradient at $t = 1$ | $\mathcal{O}(BQ^d \log Q)$ | $\mathcal{O}(BQ^d)$ | $\mathcal{O}(BQ^d)$ | $\mathcal{O}(BQ^d)$ |
| (iii). Backward shooting | $\mathcal{O}(BTq^d \log q)$ | $\mathcal{O}(BTQ^d \log Q)$ | $\mathcal{O}(BTq^d)$ | $\mathcal{O}(BTQ^d)$ |

For steps (i) and (iii), the complexity of the existing methods for computing diffeomorphisms in the high-dimensional image space is $\mathcal{O}(BTQ^d \log Q)$; in contrast, it has been shown that the complexity of Geo-SIC is $\mathcal{O}(BTq^d \log q)$. For step (ii), we convert the transformation into the spatial domain via FFT ($\mathcal{O}(BQ^d \log Q)$). We consider steps (i) and (iii) computationally dominant along with the integration of time-dependent transformation fields, and step (ii) is a one-time computation at the fixed time point $t = 1$.

Our algorithm reduces the overall complexity from $\mathcal{O}(BTQ^d \log Q)$ to $\mathcal{O}(BTq^d \log q)$ and memory consumption from $\mathcal{O}(BTq^d)$ to $\mathcal{O}(BTQ^d)$, where $q \ll Q$ and $Q$ lies in a high-dimensional imaging domain (e.g., a brain MRI with $256^3$ volxes). Please refer to our experimental section for a comparison of the run time and memory consumption between Geo-SIC and Lagomorph.