# OpenReview forum: "Geo-SIC: Learning Deformable Geometric Shapes in Deep Image Classifiers"
_NeurIPS.cc/2022/Conference — NeurIPS 2022 Accept_

### Official Review · Reviewer_GhJk · 2022-06-21

**Rating:** 7
**Confidence:** 3
**Soundness:** 3 good
**Presentation:** 3 good
**Contribution:** 4 excellent

**Summary:**

The article presents an approach to image classification using deformable atlases. This not only improves the classification itself but also makes it possible to better explain the decisions made by the models. In addition, the authors show that this approach is computationally less expensive than some previous approaches.
For the introduction of the atlas information into the deep learning model, they first train an encoder-decoder capable of predicting the atlas from the input images. In parallel, with the embeddings generated by the first one, they train a second network that will be in charge of making the predictions.


**Questions:**

*.- Improve the explanation of the equations used to build the loss functions. Include them in the annexes if you need extra space.
*.- References to Tables 1 and 2 are wrong.
*.- A comparison of the computational cost between the base models (AlexNet, CNN, ...) and the models with Geo-SIC should be presented.
*.- Due to the fact that the CNN base model is not a standard one, a better description of it should be added.


**Limitations:**

The limitations of the model are correctly addressed.



**Strengths And Weaknesses:**

Strengths:
*.- It appears to be a robust approach.
*.- For similar (or somewhat better) results to previous versions it has a lower computational cost.

Weaknesses:
*.- Equations (1-4) (basic for the construction of the loss functions) would require a somewhat more detailed explanation to facilitate the comprehensibility of the proposal. (Maybe it could be added in the annexes).
*.- The models are not sufficiently explained. It seems that there are two loss functions that are applied alternatively depending on the model being trained. It is also not clear whether the error propagates to all models or only to the one in the phase currently being trained, freezing the rest.

---

> ### Author Response · Authors · 2022-08-01
> **Response for Reviewer 3**
>
> We thank R3 for all the positive comments and constructive feedback. We will add (i) more details of the geometric learning (atlas building-based) loss function in the supplementary material, (ii) add descriptions of the CNN model parameters, and (iii) clarify that the network error propagates iteratively in the phase currently being trained in our revised manuscript. According to R3’s question of computational cost, we kindly point out that the main difference is the additional geometric learning network developed in Geo-SIC, which is included in the right panel of Fig. 6.
>
> All typos will be corrected in the revised manuscript.

---

### Official Review · Reviewer_EPtR · 2022-06-30

**Rating:** 7
**Confidence:** 5
**Soundness:** 3 good
**Presentation:** 3 good
**Contribution:** 3 good

**Summary:**

This paper presents a deep learning approach that improves the classification performance using shape priors that are jointly trained with the classification learning task without the need for pre-extracted shapes.  The core assumption is that class-specific shape information can be encoded by matching the input image to a representative atlas, in this paper this atlas is learned. The efficacy of the proposed method is showcased on synthetic and real medical data.


**Questions:**

- How would the proposed method be used for / adapted to point-based shape models?
- Are the images (training and testing) assumed to be roughly aligned? how is this accomplished? how sensitive is the method wrt misalignments in training? and how misalignment in testing can be handled?
- The learned multiple atlases of the brain data are not shown.  In figure 6, only one atlas is shown. Shouldn’t we expect an atlas per class?
- It is not clear if the heat maps in Figure 5 show shape-specific changes that are indicative of dementia, Geo-SIC is different from others, however, there is no discussion (or detailed figures) provided to demonstrate the specificity of these heatmaps to the dementia problem.

**Limitations:**

No. Authors are encouraged to discuss the limitations of the methods and delineate potential negative societal impact.

**Strengths And Weaknesses:**

Strengths:
- Jointly learning atlases and sample-specific shape representation to boost classification performance is a novel idea.
- The use of deformation-based representation allows for detailed shape descriptors and avoids pre-extracting shapes from images.
- Implicit incorporating shape information within image classifiers should improve the robustness of the classifiers to intensity variations.
- The method provides explicit classification interpretations in terms of geometric features that derive the classification task.
- The method shows classification improvements in addition to interpretability.

Weaknesses:
- Although claimed, it is not clear how would the proposed method be generalized to other representations (e.g. point-based models).
- The robustness to variations in image intensity is not demonstrated in the experiments.
- The method seems to only operate on aligned samples.

---

> ### Author Response · Authors · 2022-08-01
> **Response for Reviewer 2**
>
> We thank R2 for the positive comments and suggestions. Please see our responses to all the questions below:
>
> *Since Geo-SIC is focusing on deriving the features of geometric changes/deformations between objects in images, the data metric could be on images (as what we present in the paper), or point-based data (once the deformed model point is mathematically formulated).
>
> *Yes, we carefully removed the affine transformations on all image data, which is typically done for deformation-based models. We have not had any issues of sensitivity empirically so far.
>
> *We will show the atlases of brains for each class in the revised manuscript.
>
> *Thanks for this great point! The heatmaps in Fig. 5 are initial results of Geo-SIC and suggest that our network gains better attention on identified AD regions (such as ventricles, hippocampus, etc.) than the baselines. Our next step is to develop further analysis (i.e., utilizing feature gradient flows) to quantify and better interpret shape-specific changes particularly on brain regions that are indicative of dementia. We hope to include this work in an extended journal paper.
>
> *Great question on the robustness of Geo-SIC to the variations of image intensities! We designed a simple test by adding universal adversarial noises in the google draw images (with five classes) and found out Geo-SIC consistently achieves better accuracy (~10% higher) than classifiers that focus on image intensities. We will try to add this interesting experiment to our revised manuscript if the space allows us to.

---

### Official Review · Reviewer_TVNU · 2022-07-12

**Rating:** 6
**Confidence:** 4
**Soundness:** 3 good
**Presentation:** 3 good
**Contribution:** 2 fair

**Summary:**

This paper proposes to solve a classification task by jointly solving a problem of atlas building for each specific class. Thus the classifier network uses the shape features that are used to deform the templates as well as the features learned directly by the classifier. The trained model exhibits good classification performances as well as a good representation of the data (sharp templates).

**Questions:**

-The authors should comment on the use of Unet (including skip connections) as opposed to (variational) autoencoders in order to generate the shape features.

- How is the proposed approach sensitive to the rigid transform between the template and image ? How is this taken into account in the proposed framework ?

- At inference time, the choice of the template to be towards the image is unknown. How to combine the classification network and the shape reconstruction network to get  the registered  template ?

- The authors should try to provide explanations about the slight improved performance for the generation of atlas since the atlas generation method is similar to the ones proposed before

**Ethics Review Area:**

["I don’t know"]

**Limitations:**

- The authors do not provide a set of potential limitations of their work.
- Limitations of the work include :
i) the proposed approach is only valid for same modality images since simple similarity criteria (SSD) are used
ii) It represents shape as the deformation of a single template which is only valid for a simple unimodal distribution of shapes
iii) it does not allow to estimate the template deformation at inference time


**Strengths And Weaknesses:**

Strengths : the paper is well written , with solid mathematical basis and the proposed concept of joint learning is simple to grasp. The results are solid with extensive comparisons with baseline methods in various aspects : classification performance, as well as representation by deformable templates Two datasets are chosen, the former being toy examples and the latter being related to neuroimaging MRIs.

Weaknesses : the main contribution is to solve jointly two tasks instead of solving them sequentially (two step method). This makes it a fairly limited  contribution for this conference. Another weakness is that during inference, the choice of the atlas is not known and therefore comparison of the input image with the deformed atlas cannot be done. Besides, the paper assumes that each class can be well represented by the deformation of a single template (unimodal distribution)

---

> ### Author Response · Authors · 2022-08-01
> **Response for Reviewer 1**
>
> We thank the comments and suggestions from R1. Please see our responses to all questions below:
>
> *Aside from developing a joint optimization of geometric learning (based on atlas building) and classification, our proposed Geo-SIC has other major contributions that have been rarely explored in the literature. As R2&R3 also pointed out, Geo-SIC was the first to learn deformation-based geometric descriptors within an image classifier. It provides an image distance function of both intensity and geometric changes that are most relevant to classify different groups. In contrast to previously used two-step approaches, Geo-SIC has multiple advantages, such as (i) a direct optimization of geometric feature learning and group classification; (ii) a reduced computational cost of training inference by employing a low-dimensional parameterization of deformation fields in a bandlimited space; and (iii) an improved performance of model accuracy and robustness. We will make sure to clarify these contributions in the introduction section.
>
> *The current work of Geo-SIC focuses on deriving geometric features that measure the differences between two objects; we carefully removed affine (rigid) transformations for all datasets. We agree that it would be interesting to see how rigid transformations play a role in the classification task. This could be considered as an extension on top of our current work.
>
> *At inference time, our boosted classifier with learned parameters in both image and geometric feature space will predict the label of each testing image. While a registered template is not necessary for the testing, a user can pass the predicted image with a label (and revealed atlas) to the geometric learning network to predict transformations (to generate a deformed atlas).
>
> *Thanks for bringing this up! We will add an elaborated discussion on why our atlas building has improved performance over the current approaches. To summarize, our model Geo-SIC parameterized the high-dimensional deformation fields in a nicely-investigated low-dimensional bandlimited space [44], which provides faster convergence with superior accuracy due to the local minima issues in the original high-dimensional space.
>
> *We thank R1 for pointing out the potential limitations. We will include a paragraph with regard to broader options for the image similarity metric, an extension to multiple-template within each class data, etc., in the revised manuscript.

---

### Meta-Review · Area_Chair_HLGb · 2022-08-26

**Recommendation:** Accept
**Confidence:** Certain

**Metareview:**

Although there were a couple of initial questions/concerns about certain aspects of the paper, all reviewers appreciated the approach, the quality of presentation and the empirical results. After reading all responses by the authors, my impression is that all questions have been answered satisfactorily during the rebuttal period. Hence, I do recommend acceptance of this paper.

**Award:**

No

---

### Decision · Program_Chairs · 2022-09-14

Accept